# Mechanism of Action of Methotrexate Against Zika Virus

**DOI:** 10.3390/v11040338

**Published:** 2019-04-10

**Authors:** Sungjun Beck, Zhe Zhu, Michelli F. Oliveira, Davey M. Smith, Jeremy N. Rich, Jean A. Bernatchez, Jair L. Siqueira-Neto

**Affiliations:** 1Skaggs School of Pharmacy and Pharmaceutical Sciences, University of California, San Diego, La Jolla, CA 92093, USA; sjbeck@ucsd.edu; 2Sanford Consortium for Regenerative Medicine, La Jolla, CA 92093, USA; zhz504@ucsd.edu (Z.Z.); drjeremyrich@gmail.com (J.N.R.); 3Department of Medicine, Division of Regenerative Medicine, School of Medicine, University of California, San Diego, La Jolla, CA 92093, USA; 4Department of Medicine, Division of Infectious Diseases and Global Public Health, University of California, San Diego, La Jolla, CA 92093, USA; mfariadeoliveira@ucsd.edu (M.F.O.); d13smith@ucsd.edu (D.M.S.); 5Veterans Affairs San Diego Healthcare System, San Diego, CA 92093, USA; 6Center for Discovery and Innovation in Parasitic Diseases, University of California, San Diego, La Jolla, CA 92093, USA

**Keywords:** Zika virus, antivirals, nucleotide metabolism, methotrexate, dihydrofolate reductase

## Abstract

Zika virus (ZIKV), which is associated with microcephaly in infants and Guillain-Barré syndrome, reemerged as a serious public health threat in Latin America in recent years. Previous high-throughput screening (HTS) campaigns have revealed several potential hit molecules against ZIKV, including methotrexate (MTX), which is clinically used as an anti-cancer chemotherapy and anti-rheumatoid agent. We studied the mechanism of action of MTX against ZIKV in relation to its inhibition of dihydrofolate reductase (DHFR) in vitro using Vero and human neural stem cells (hNSCs). As expected, an antiviral effect for MTX against ZIKV was observed, showing up to 10-fold decrease in virus titer during MTX treatment. We also observed that addition of leucovorin (a downstream metabolite of DHFR pathway) rescued the ZIKV replication impaired by MTX treatment in ZIKV-infected cells, explaining the antiviral effect of MTX through inhibition of DHFR. We also found that addition of adenosine to ZIKV-infected cells was able to rescue ZIKV replication inhibited by MTX, suggesting that restriction of de novo synthesis adenosine triphosphate (ATP) pools suppresses viral replication. These results confirm that the DHFR pathway can be targeted to inhibit replication of ZIKV, similar to other published results showing this effect in related flaviviruses.

## 1. Introduction

Mosquito-borne flaviviruses, such as Dengue virus (DENV), Yellow Fever virus (YFV), West Nile virus (WNV), Japanese Encephalitis virus (JEV), and Zika virus (ZIKV), are well characterized as human pathogenic viruses [1]. Recently, ZIKV has become a global public health threat that has brought substantial economic burden to affected countries during the recent outbreak in Latin America [2]. Known for its unique neurotropism, ZIKV was associated with a substantial number of ZIKV-induced neurological disorders, mainly microcephaly in infants and Guillain-Barré syndrome, during 2015 and 2016 in Brazil and Colombia [3,4]. Currently, there are no specific antivirals or vaccines to treat ZIKV infection.

High-throughput screening (HTS) of various compound libraries against ZIKV have been published, and several small molecules active against ZIKV were identified [5,6,7,8]. One active compound was methotrexate (MTX), used to treat a variety of diseases including leukemia, psoriatic arthritis, and rheumatoid arthritis [9,10,11]. As a chemotherapy agent, the main function of MTX is to antagonize dihydrofolate reductase (DHFR) to decrease de novo synthesis of purines and pyrimidines [12,13], thereby inhibiting the viability of highly replicating cells. Also, MTX has been reported to interfere with a variety of cellular mechanisms, such as oxidative stress or cellular differentiation via methylation [14,15].

Interestingly, DHFR inhibitors have also been investigated to treat infectious diseases. Due to its formation of a specific dimer with thymidylate synthase (TS) [16], protozoan DHFR has been studied as a drug target to treat parasitic diseases [17]. As antiviral agents, MTX and the antimetabolite floxuridine have been shown to decrease the replication of DENV through inhibition of host DHFR and TS enzymes [18]. Here, we demonstrate the antiviral activity of MTX against ZIKV and probe its mechanism of action using downstream metabolites of the DHFR pathway, leucovorin and adenosine.

## 2. Materials and Methods

### 2.1. Cell and Virus Culture

Vero cells (ATCC CCL-81) were purchased from ATCC (Manassas, VA, USA) and cultured in Dulbecco’s modified eagle medium (DMEM) (Thermo-Fisher, Waltham, MA, USA) with high glucose (4.5 g/L), 10% fetal calf serum (FCS) (Sigma-Aldrich, St. Louis, MO, USA), and 1% Penicillin-Streptomycin (PS) (Sigma-Aldrich, St. Louis, MO, USA). Human neural stem cells (hNSCs) (Y40050) were purchased from Clontech and cultured in either Neurobasal-A medium with phenol red (Thermo-Fisher, Waltham, MA, USA) or without phenol red (Thermo-Fisher, Waltham, MA, USA) with the addition of B27 supplement (1:100, Thermo Fisher, #12587010), N2 supplement (1:200, Invitrogen, Carlsbad, CA, USA, #17502-048), 20 ng/mL fibroblast growth factor (FGF) (R&D Systems, Minneapolis, MN, USA, 4114-TC-01M), 20 ng/mL EGF (R&D Systems, Minneapolis, MN, USA, 236-EG-01M), GlutaMax (Thermo-Fisher, Waltham, MA, USA, #35050061), and sodium pyruvate. Both Vero and hNSCs were cultured in T75 flasks (Corning, Corning, NY, USA). Two strains of ZIKV from Puerto Rico (PRV) (PRVABC59, NR-50240) and Panama (H/PAN) (H/PAN/2016/BEI-259634, NR-50210), were cultured on Vero cells in DMEM with high glucose (4.5 g/L), 1% fetal bovine serum (FBS) (Sigma-Aldrich, St. Louis, MO, USA), and 1% PS (Sigma-Aldrich, St. Louis, MO, USA). When 100% confluency of Vero cells was reached, the initial medium was removed, and the cells were infected with ZIKV at multiplicity of infection (MOI) of 0.5. At 24 h post-infection (PI), the initial medium was removed, and fresh medium was added. At 48 h PI, the medium was collected and spun down at 1500 rpm for 10 min. The supernatant was collected, 1% DMSO was added, and the sample was stored at −80 ℃. The titer of the virus stock was measured by plaque assay.

### 2.2. Plaque Assay

Vero cells were seeded at a density of 30,000 cells/well on a 96-well plate (Corning, Corning, NY, USA) and incubated at 37 °C and 5% CO_2_ for 24 h before infection. Each ZIKV sample was diluted by 10-fold serial dilution, added to Vero cells in at least duplicate, then incubated for 1 h. After incubation, media with virus dilutions were removed from the cells, and ZIKV-infected Vero cells were covered with an overlay consisting of DMEM with 0.35% UltraPure agarose (Thermo-Fisher, Waltham, MA, USA). The plates were further incubated for 72 h and fixed with 37% formaldehyde overnight. After fixation, the agarose overlay was aspirated, and fixed cell monolayer was stained with 0.25% crystal violet. For each sample, the titer of ZIKV was reported in PFU/mL.

### 2.3. Immunofluorescence Imaging

Vero cells were seeded at a density of 30,000 cells/well in a black 96-well plate (Corning, Corning, NY, USA) and incubated at 37 °C and 5% CO_2_ for 24 h before infection. Then, Vero cells were infected with ZIKV at MOI of 0.2. The infected Vero cells were incubated for 48 h then fixed with 4% formaldehyde in phosphate buffered saline (PBS). After fixation, the cells were permeabilized with 0.2% Triton X-100 in PBS for 5 min, then blocked with 1% bovine serum albumin (BSA) (VWR, Radnor, PA, USA) in PBS for 30 min at room temperature. After blocking, the cells were incubated with diluted primary antibody, anti-flavivirus group antigen primary antibody, clone D1-4G2-4-15 (MilliporeSigma, Burlington, MA, USA, #MAB10216), in PBS with 1% BSA and 0.1% Triton X-100 at 4 °C overnight. The cells were washed with PBS three times and treated with diluted secondary antibody, goat anti-mouse Alexa Fluor 594 (Thermo-Fisher, Waltham, MA, USA, #A-11032), in PBS with 1% BSA for 1h at room temperature while covered with aluminum foil. The cells were then washed with PBS three times and stained with SYTOX Green (Thermo-Fisher, Waltham, MA, USA, #R37168) to visualize nuclei. To determine MTX’s effect on decreasing ZIKV envelope protein immunostaining, 5 μM MTX was added to ZIKV-infected Vero cells under the same conditions as Immunofluorescence images of the cells were acquired using either a ZEISS Fluorescence Microscope (Carl Zeiss, Jena, Germany, Axio Vert A1) or ImageXpress Micro Fluorescence Microscope (Molecular Devices, San Jose, CA, USA). Images were analyzed using Zen2 Software (Carl Zeiss, Jena, Germany), MetaXpress (Molecular Devices, San Jose, CA, USA), and ImageJ (https://imagej.nih.gov/ij/).

### 2.4. Cytotoxicity and Efficacy Study of MTX

The cytotoxic concentration 50 (CC_50_) of MTX (Sigma-Aldrich, St. Louis, MO, USA, #M9929) was determined using two reagents, CellTiter-Glo^®®^ (CTG) (Promega, Madison, WI, USA, #G7570), measuring ATP levels as a readout for cell viability, and CellTiter-Fluor™ (CTF) (Promega, Madison, WI, USA, #G6080), measuring live-cell protease activity, as per manufacturer’s protocol. To validate the CC_50_ value from the two cell viability assays, a direct cell number count was performed using trypan-blue staining after MTX treatment. The efficacy of MTX against ZIKV was studied by reduction of ZIKV titer using four different MTX concentrations, 50.0 µM, 6.25 µM, 0.781 µM, and 0.0977 µM, in dose-response manner. The effective concentration 50 (EC_50_) of MTX was determined by immunofluorescence imaging. With Vero cells, a mixture of Vero cells at a density of 3000 cells/well and ZIKV inoculum (MOI of 0.2) was prepared and seeded in a black 384-well plate (Greiner Bio-One, Kremsmünster, Austria) with MTX in a dose-response manner. For hNSCs, a density of 10,000 cells/well was seeded in a black 96-well plate (Corning, Corning, NY, USA) with Geltrex (Thermo-Fisher, Waltham, MA, USA, A1413203) coating, following the manufacture’s protocol. Once the hNSCs formed a monolayer, the cells were infected with ZIKV (MOI of 0.2) and with MTX in a dose-response manner. For antibody staining, the same protocol was followed as described above. For EC_50_ analysis, ImageXpress (Molecular Devices, San Jose, CA, USA) built-in imaging modules were used to detect ZIKV envelope proteins, either measuring the average cytoplasm fluorescence intensity or the number of infection-positive cells per total nuclei scoring (DAPI staining). For both CC_50_ and EC_50_ calculations, PRISM8 (GraphPad, La Jolla, CA, USA) software was used.

### 2.5. Rescue of ZIKV Replication after MTX Treatment by Leucovorin

To test the rescue effect of leucovorin on ZIKV replication with MTX treatment, Vero cells and hNSCs were seeded in a 96-wells plate (Corning, Corning, NY, USA) at a cell density of 30,000 and 10,000 cells/well, respectively. The Vero cells and hNSCs were then infected with ZIKV at MOI 0.2 and 0.1, respectively, and simultaneously treated with 5 μM MTX, or a combination of 5 μM MTX with 50 μM folic acid (Sigma-Aldrich, St. Louis, MO, USA, #F7876) or leucovorin (Sigma-Aldrich, St. Louis, MO, USA, #F7878). The cells were further incubated for 48 h at 37 °C and 5% CO_2_. After incubation, the supernatant was collected to measure the virus titer. The cells were then treated under the same conditions as described above but without ZIKV infection to assess the effect of leucovorin on countering the cytotoxic effects of MTX on the cells. Then, the cell viability was measured using CTG (Promega, Madison, WI, USA) reagent. To control for the effect of leucovorin on host cell viability, 50 μM of leucovorin was added to both Vero cells and hNSCs under the same conditions using CTG (Promega, Madison, WI, USA) reagent.

### 2.6. Rescue Effect of GAT Medium on ZIKV Replication and Cell Viability from Methotrexate

To test the rescue effect of GAT medium (Glycine, Adenosine, and Thymidine, Sigma-Aldrich, St. Louis, MO, USA, #410225, #A4036 and #T1895, respectively) on MTX-mediated suppression of ZIKV replication, Vero cells were seeded in a 96-well plate at a cell density of 10,000 cells/well. GAT medium was prepared as described previously [19], in MEM (Thermo-Fisher, Waltham, MA, USA) with 10% FCS (Sigma-Aldrich, St. Louis, MO, USA) and 1% PS (Sigma-Aldrich, St. Louis, MO, USA) with a final concentration of 0.67 mM glycine, 37.5 μM adenosine, and 41.3 µM thymidine. Vero cells were infected with ZIKV at a MOI of 0.2 and simultaneously treated with 5 μM MTX in GAT medium. The cells were further incubated for 48 h at 37 °C and 5% CO2. After incubation, the supernatant was collected to measure the virus titer. Vero cells were treated under the same conditions as described above but without ZIKV infection as a cell viability control. Cell viability was measured using CTG reagent (Promega, Madison, WI, USA). To investigate the rescue effect of individual components of GAT medium, Vero cells were infected with ZIKV as described above. Then, a combination of 5 μM MTX with 0.67 mM glycine, 37.5 μM adenosine, 41.3 µM thymidine, or both adenosine and thymidine were simultaneously added to the infected Vero cells. The treated Vero cells were further incubated for 48 h, and supernatant from each sample was collected to measure the virus titer. The same conditions were also applied to Vero cells without ZIKV infection to study which component of GAT medium could protect cells from MTX-induced cytotoxicity, measure by cell viability using both CTG and CTF reagents. To control for the effect of GAT medium and adenosine on host cell viability, GAT medium and 37.5 μM adenosine were added to Vero cells uder the same conditions using CTG (Promega, Madison, WI, USA) and CTF (Promega, Madison, WI, USA) reagents.

## 3. Results

### 3.1. Inhibition of ZIKV Replication with MTX

To validate the antiviral effect of MTX against ZIKV, immunofluorescence imaging and plaque assay were performed. Through immunofluorescent detection, reduced ZIKV (H/PAN MOI 0.2) envelope protein synthesis in Vero cells was observed from 5 μM MTX treatment at 48 h PI (Figure 1A). The fluorescence signal difference was also measured between 5 μM MTX treatment and DMSO control (Figure 1E). We next tested the replication kinetics of ZIKV (H/PAN and PRV; both MOI 0.2) in Vero cells by plaque assay at three different time points, 1 h, 48 h, and 96 h PI (Figure 1B,C). Upon 5 μM MTX treatment, the replication of two ZIKV isolate strains had greatest reduction (approximately 10-fold decrease in ZIKV titer) at 48 h PI, which is consistent with the immunofluorescence images. However, the ZIKV replication resumed at 96 PI, compared to DMSO control, indicating MTX could not continuously suppress ZIKV virion formation after 48 h PI. In the case of efficacy of MTX against ZIKV, four different MTX concentrations, 50.0 µM, 6.25 µM, 0.781 µM, and 0.0977 µM, were incubated with ZIKV-infected Vero cells (H/PAN MOI 0.2) (Figure 1D). At 48 h PI, the ZIKV titer decreased about 10-fold at 6.25 µM MTX treatment. To confirm the results from the plaque reduction data, a dose-response of MTX added to infected Vero cells showed reduction in the ZIKV fluorescence signal from ZIKV (H/PAN MOI 0.2) at 48 h PI (Figure 1F); and was used to calculate the EC_50_ value of MTX against ZIKV.

### 3.2. Cell Cytotoxicity and Antiviral Efficacy of Methotrexate

To study the cytotoxicity of MTX, we used CTG and CTF reagents for cell viability assays. We tested the compound at different concentrations to calculate the CC_50_ of MTX in the two host cells Using CTG reagent, the CC_50_ of MTX in Vero cells and hNSCs were calculated to be 0.104 μM and 0.0163 μM, respectively (Table 1). However, using CTF reagent, the CC_50_ of MTX in Vero cells and hNSCs were calculated to both be >100 μM. The two CC_50_ values showed drastic variance. Accordingly, to validate which CC_50_ value is more appropriate, the number of Vero cells were directly counted using microscopy with trypan-blue staining after 0.5 μM of MTX treatment (Figure 2A,B). At an initial cell density of 10,000 cells/well, 0.5 μM MTX treatment showed static cell density of Vero cells after 48 h incubation comparable to that of the DMSO control. However, at an initial cell density of 30,000 cells/well, both 0.5 μM and 5 μM of MTX showed similar cell viability when using CTF reagent. For the EC_50_ calculation, immunofluorescence images of ZIKV infected (H/PAN MOI 0.2) Vero cells and hNSCs with different concentrations of MTX in dose-response manner were taken. Reduction in ZIKV fluorescence signal per nuclei score was used to calculate the EC_50_ of MTX for infected Vero cells and hNSCs, which were 0.245 μM and 0.334 μM, respectively (Table 1).

### 3.3. Relationship between MTX, Folate, and ZIKV Replication

To understand the mechanism of action for the antiviral effect of MTX, we proceeded to probe the dihydrofolate reductase (DHFR) pathway. DHFR, which reduces dihydrofolate (DHF) to tetrahydrofolate (THF), is a key enzyme for de novo synthesis of purines and thymidylate, thereby playing a critical role in cellular growth. As an antifolate, MTX has been reported as a competitive inhibitor of DHFR [20] and shares a similar chemical structure to folate, the natural substrate of DHFR. Through MTX-mediated antagonism of DHFR, folate metabolism is inhibited, which in turn decreases cellular replication. However, addition of leucovorin (folinic acid), a downstream metabolite of the DHFR pathway, can readily rescue the antagonistic effect of MTX in cancer cells [21].

To understand the metabolic relationship between MTX, folate, and leucovorin, Vero cells were infected with two ZIKV isolate strains, H/PAN and PRV (both MOI 0.2) and treated with small molecules under three different conditions: MTX, MTX with folic acid, and MTX with leucovorin. At 48 h PI, the combination treatment of MTX with folic acid could not rescue the ZIKV replication; however, the combination treatment of MTX with leucovorin rescued ZIKV replication, compared to DMSO control (Figure 3A,B). To study the effect of these three different conditions on Vero cells, the same experiment was performed with the uninfected Vero cells. Using CTG reagent, we measured the cell viability of Vero cells at each condition (Figure 3D). A total of 50 µM leucovorin could rescue the cell viability of Vero cells after MTX treatment. Furthermore, the same rescue effect for leucovorin on ZIKV replication and cell viability during MTX treatment was observed in hNSCs (Figure 3C,E). To study if leucovorin enhances the viability of the host cells, 50 µM leucovorin was added to Vero cells and hNSCs. No significant difference in cell viability was observed using CTG reagent (Figure 3F,G). These results support a model where the decrease in ZIKV titer from MTX treatment occurs through antagonism of DHFR.

### 3.4. Rescue Effect of GAT Medium on ZIKV Replication and Cell Viability during MTX Treatment

Based on our evidence that the ZIKV plaque-reducing effect of MTX occurs through antagonism of DHFR, we next assessed which of the metabolites involved in purine and pyrimidine synthesis could affect ZIKV replication. Previous reports have indicated that the cytotoxicity of MTX was reversed by not only leucovorin, but also by GAT medium, which contains glycine, adenosine, and thymidine, in human fibroblasts [19]. We therefore studied the rescue effect of GAT medium and each individual component of the medium on ZIKV replication during MTX treatment in Vero cells. GAT medium could rescue both cell viability and ZIKV replication (H/PAN MOI 0.2) during MTX treatment (Figure 4A,B). Subsequently, glycine, adenosine, thymidine, and a combination of adenosine and thymidine were individually added with MTX to ZIKV infected Vero cells. Interestingly, only adenosine rescued ZIKV replication during MTX treatment (Figure 4C). To observe if adenosine alone could rescue cell viability during MTX treatment, the cell viability of Vero cells was studied using both CTG and CTF reagents. Whereas glycine and thymidine could not rescue the cell viability, adenosine alone could, in both CTG and CTF assays. (Figure 4D,E). Lastly, to study if GAT medium and adenosine affects the viability of Vero cells, the same concentration of GAT medium and adenosine were added to Vero cells under identical conditions as those described above using CTG and CTF reagents as readouts. Both GAT medium and adenosine did not alter the viability of Vero cells (Figure 4F,G).

## 4. Discussion

In this study, we validated MTX as an antiviral targeting ZIKV replication. MTX was previously discovered as a potent hit molecule against ZIKV in human brain microvascular endothelial cells, with inhibitory concentration 50 (IC_50_) of 0.28 μM and host cytotoxicity (CC_50_) of >10 μM [5]. It is well known that folic acid supplementation during pregnancy reduces the risk of neural tube defects and other neurodevelopmental complications of pregnancy. It is therefore interesting that inhibition of this pathway prevents ZIKV replication, suggesting that the neurodevelopmental effects of ZIKV and folic acid deficiency may proceed through similar mechanisms. MTX is approved by the FDA to treat diseases such as cancer and rheumatoid arthritis. Accordingly, the mechanism of action of MTX against such diseases have been well studied. Primarily, MTX is known to antagonize DHFR as a competitive inhibitor [20] and antagonized DHFR causes inhibition of de novo synthesis of purines and pyrimidines, which are essential for cell replication [22]. Hence, MTX shows great cytotoxicity to highly replicating cancer cell lines. Interestingly, with 5 μM MTX treatment, reduced ZIKV replication was observed from infected Vero cells and hNSCs. Although MTX was able to reduce ZIKV titers about 10-fold for 48 h PI, viral levels began to increase past this time point. Observing the antiviral effect of MTX against ZIKV, the toxicity of MTX in Vero cells and hNSCs were measured. Understanding that MTX can inhibit purine and pyrimidine metabolism, cells treated with MTX showed significantly decreased viability in terms of ATP level. Accordingly, CC_50_ measurement with CTG reagent also resulted in high cytotoxicity in the two host cells; however, such toxicity could be misleading because the drug mechanism directly interferes with ATP levels, i.e., the readout measurement of the assay. Accordingly, Vero cells with 0.5 μM MTX treatment, exceeding the CC_50_ value obtained from CTG reagent, did not exhibit phenotypic cytotoxicity but rather showed slower cellular replication compared to untreated controls. Furthermore, direct quantification of viral stain per cell revealed a substantial decrease in viral particles per cell for the MTX-treated groups compared to DMSO vehicle controls (Figure 1 and Table 1).

MTX decreased ZIKV titer in Vero cells and hNSCs by antagonizing DHFR. The inhibited DHFR activity by MTX was readily reversed by simultaneous co-treatment with leucovorin, thereby rescuing ZIKV replication from MTX as well. Further metabolite analysis with GAT medium allowed us to understand adenosine alone could rescue ZIKV replication by increasing ATP level from MTX treatment. Although MTX is known to inhibit TS [23], thymidine alone could not rescue the ZIKV replication. Considering the genome of ZIKV, a single-stranded positive sense RNA (ssRNA+), thymidine triphosphate (TTP) would not be incorporated into its genome during the replication [13]. In fact, Fischer et al. reported inhibition of TS can reduce flavivirus replication but also found DENV replication was not rescued when excess amount of thymidine was co-treated with MTX, suggesting an alternative mechanism such as activation of P53 for general antiviral activity due to low levels of thymidine [18].

Together, our results identified that the DHFR pathway can be targeted for the development of inhibitors of ZIKV and other flaviviruses, as well as Chikungunya virus that shares similar features to those of ZIKV [24]. Clearly, repurposing MTX as an antiviral against ZIKV infection would not be an ideal option for pregnant women, since MTX has been used as a clinical abortion agent [25]. However, MTX could conceivably be used in patients that would otherwise have limited treatment options, such as the immunocompromised. This study also acts as a proof-of-principal that the DFHR pathway can be targeted generally, and other enzymes in this pathway should be investigated as drug targets with potential for lower host toxicity.

Although excess amount of thymidine could not rescue the cell viability from MTX in vitro (Figure 4D,E), such a combination should be tested in vivo. Considering the number of reports on the rescue effect of thymidine from MTX [26,27,28], it is possible that MTX-thymidine could be an optional treatment for ZIKV infection: While MTX inhibits ZIKV replication, thymidine would rescue the host from the cytotoxicity of MTX. In conclusion, this study clarified the antiviral mechanism of MTX, and suggests the DHFR pathway may be targeted for antiviral purpose.

## Figures and Tables

**Figure 1 viruses-11-00338-f001:**
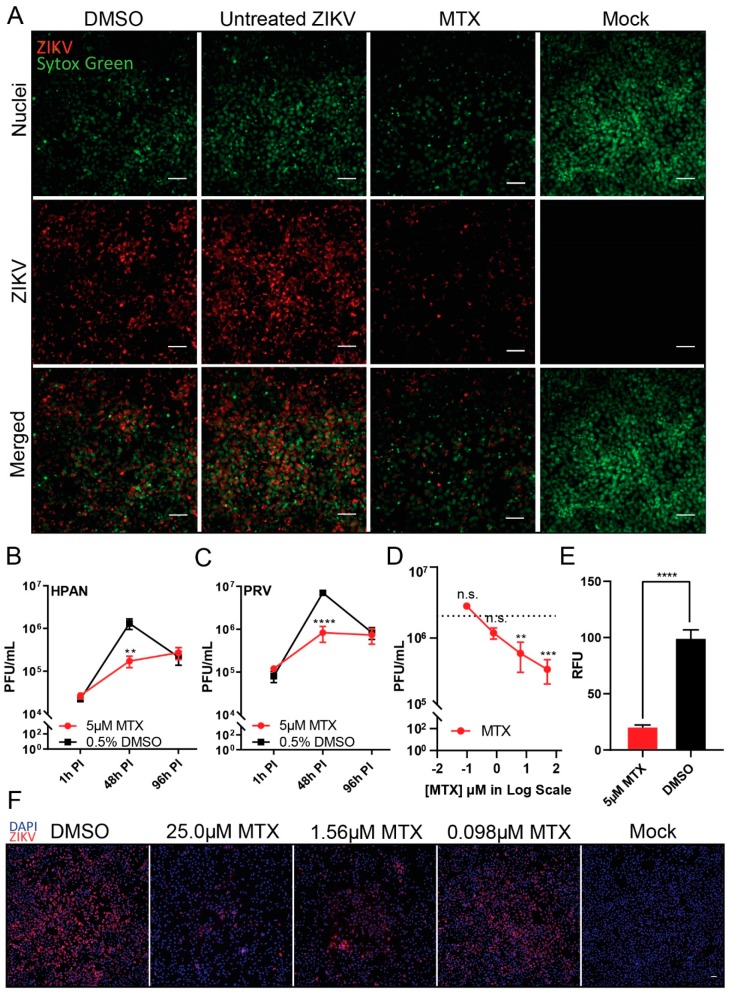
Inhibition of Zika virus (ZIKV replication in Vero Cells after methotrexate (MTX) treatment. ZIKV-infected Vero cells were treated with 5 µM MTX or 0.5% of DMSO as a negative control. (**A**) Immunofluorescence images (20X) of ZIKV-infected (H/PAN/2016/BEI-259634, NR-50210 strain (H/PAN), multiplicity of infection (MOI) 0.2) Vero cells were acquired to analyze the level of ZIKV-envelope protein after 5 µM MTX treatment at 48 h post-infection (PI). SYTOX green was used to stain nuclei of Vero cells. Scale bars represents 5 µm. The virus titer of two ZIKV strains, (**B**) H/PAN MOI 0.2 and (**C**) PRVABC59, NR-50240 strain ZIKV (PRV) MOI 0.2, were measured after 5 µM MTX treatment at three different time points, 1 h PI, 48xh PI, and 96 h PI. (**D**) ZIKV titer after MTX treatment in dose-response manner (8-fold, 50 µM, 6.25 µM, 0.781 µM, and 0.0977 µM) from ZIKV-infected (H/PAN MOI 0.2) Vero cells at 48 h PI. (**E**) Average cytoplasm fluorescence signal of ZIKV-envelope protein per nuclei from ZIKV-infected (H/PAN MOI 0.2) Vero cells with 5 µM MTX treatment was measured at 48 h PI. RFU = relative fluorescence units. (**F**) Immunofluorescence images (10×) of ZIKV-infected (H/PAN MOI 0.2) Vero cells with three different MTX concentrations (6-fold dilution series, 25.0 µM, 1.56 µM, 0.098 µM) were acquired to observe the reduction in ZIKV-envelope protein’s fluorescence signal at 48 h PI. The scale bar represents 50 µm. For (E) and (F), DAPI was used to stain nuclei of Vero cells. At least two independent replicates were performed. For (B) and (C), two-way ANOVA, followed by Sidak’s multiple comparisons test, were used for statistical analysis. For (D), one-way ANOVA, followed by Tukey’s multiple comparisons test, were used for statistical analysis; average viral titer at each MTX concentration was compared to that of untreated ZIKV at 48 h PI as a control. For (**E**), Student’s t-test was used for statistical analysis (two-tailed distribution, heteroscedastic). Error bars represent standard error of the mean (SEM). ** *p* ≤ 0.01, *** *p* ≤ 0.001, **** *p* ≤ 0.0001, n.s. not significant.

**Figure 2 viruses-11-00338-f002:**
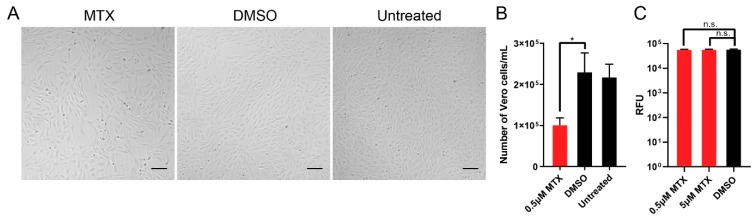
Determination of cytotoxicity of MTX. To determine the cytotoxicity of MTX, the number of Vero cells after MTX treatment was directly counted by trypan-blue staining. Then, the number of live cells were counted under a light microscope with a hemocytometer. (**A**) 10× bright-field images of Vero cells with 0.5 µM MTX treatment with initial cell density of 10,000 cells/100 µL. Scale bars represent 5 µm. (**B**) The cell density of Vero cells after 0.5 µM MTX treatment after incubation for 48 h at 37 °C and 5% CO_2_ (NuAir, Plymouth, MN, USA). (**C**) Cell viability assay using CTF reagent with initial cell density of 30,000 cells/100µL. Two MTX concentrations (5 µM and 0.5 µM) were tested. At least two independent replicates were performed. One-way ANOVA, followed by Tukey’s multiple comparisons test, were used for statistical analysis. The error bars represent the standard error of the mean (SEM). * *p* ≤ 0.05.

**Figure 3 viruses-11-00338-f003:**
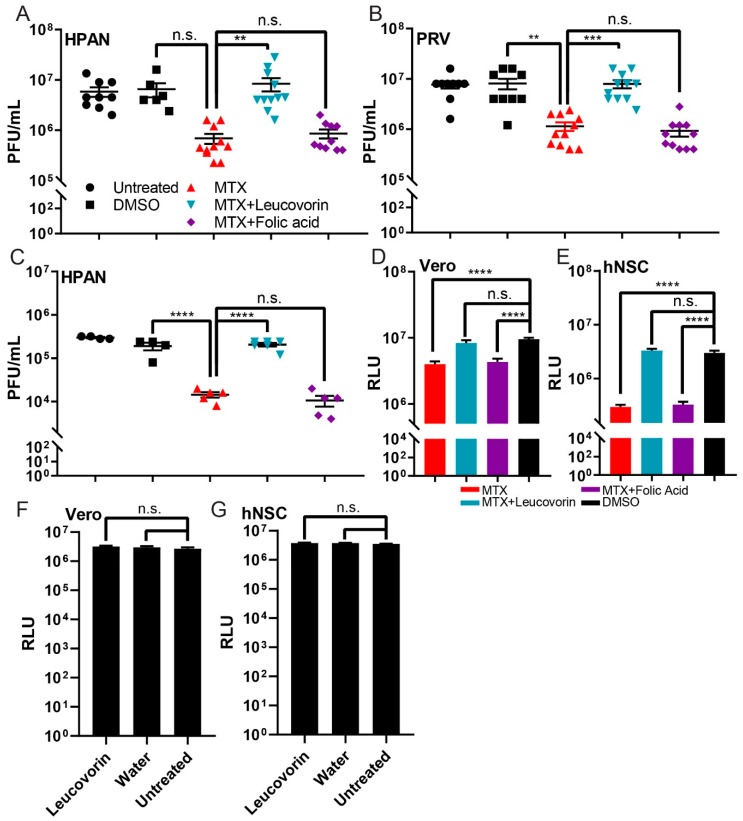
Rescue effect of leucovorin on the cell viability and ZIKV replication during MTX treatment. Two host cell lines, Vero and human neural stem cells (hNSCs), were used to examine the mechanism of action of MTX against ZIKV replication through the DHFR pathway. Virus titers of the two ZIKV strains, H/PAN MOI 0.2 (**A**) and PRV MOI 0.2 (**B**) on Vero cells were measured by standard plaque assay. With H/PAN infection, the ZIKV titer of the DMSO control was not significantly different from that of 5 µM MTX-treated sample in Vero cells (*p* = 0.11) (**C**) Virus titer of H/PAN MOI 0.1 on hNSCs was measured by standard plaque assay. (**D**) The cytotoxicity of 5 µM MTX and co-treatment of MTX with 50 µM folic acid or leucovorin on Vero cells was studied by CTG reagent. (**E**) The cytotoxicity of 5 µM MTX and co-treatment of MTX with 50 µM folic acid or leucovorin on hNSCs was cells was studied by CTG reagent. (**F**,**G**) 50 µM Leucovorin and water, as a vehicle control, were added to Vero cells and hNSCs to determine their effect on host cell viability using CTG reagent. RLU = relative luminescence units. At least two independent replicates were performed. One-way ANOVA, followed by Tukey’s multiple comparisons test, were used for statistical analysis. The error bars represent the standard error of the mean (SEM). ** *p* ≤ 0.01, *** *p* ≤ 0.001, **** *p* ≤ 0.0001, n.s. not significant.

**Figure 4 viruses-11-00338-f004:**
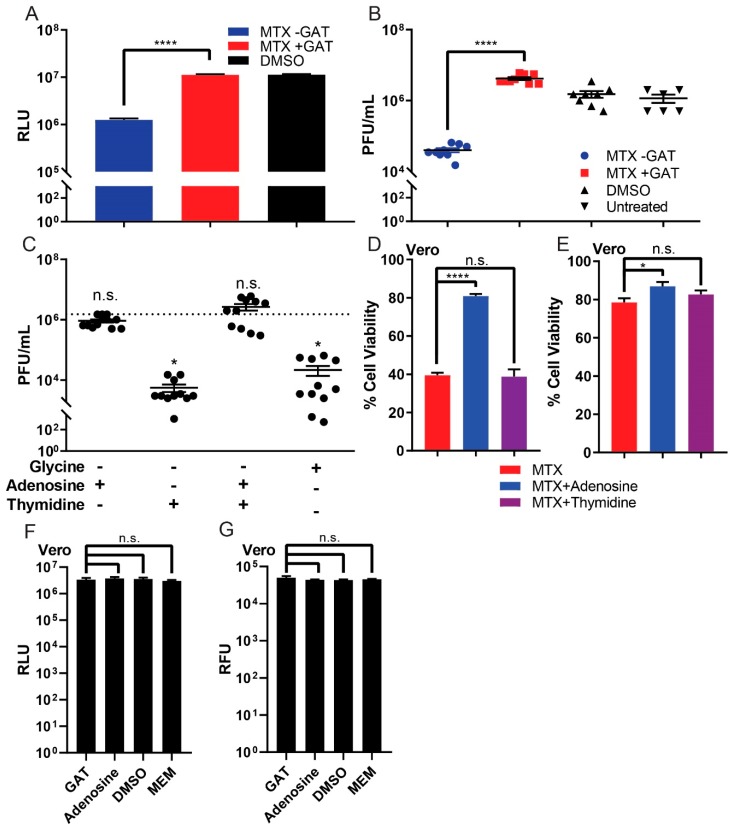
Rescue effect of GAT medium on cell viability and ZIKV replication during MTX treatment. (**A**) Cell viability of Vero cells was studied by CTG reagent after MTX treatment with or without GAT medium. (**B**) GAT medium rescued the ZIKV replication from MTX treatment of the ZIKV-infected (H/PAN MOI 0.2) Vero cells. (**C**) Adenosine alone can save ZIKV replication during MTX treatment of the ZIKV-infected (H/PAN MOI 0.2) Vero cells. (**D**) Exogenous adenosine rescued cellular ATP levels, but thymidine and glycine could not rescue ATP levels during MTX treatment in Vero cells as measured by CTG assay. (**E**) Adenosine also rescued live-protease activity measured by CTF reagent during MTX treatment in Vero cells. (**F**,**G**) GAT media and adenosine were added to Vero cells and hNSCs to study determine their effect on host cell viability using CTG and CTF reagents. RLU = relative luminescence units. At least two independent replicates were performed. The error bars represent the standard error of the mean (SEM). One-way ANOVA, followed by Tukey’s multiple comparisons test, was used for statistical analysis. * *p* ≤ 0.05, **** *p* ≤ 0.0001, n.s. not significant.

**Table 1 viruses-11-00338-t001:** Host cytotoxicity (CC_50_) and effective concentration 50 (EC_50_) values of MTX in Vero cells and hNSCs. CTG (CellTiter-Glo^®^) measures adenosine triphosphate (ATP) as a luminescent readout of cell viability biomarker. CTF (CellTiter-Fluor™) measures live protease activity as a fluorescent readout of cell viability biomarker. Immunofluorescence images of ZIKV-infected (H/PAN MOI 0.2) Vero cells and hNSCs with MTX in a dose-response manner. Average cytoplasm fluorescence signal (ZIKV envelope protein) was measured per nuclei scoring. At least two independent replicates were performed for each experiment.

**CC50 (µM) Values of MTX**		
**Assay**	Vero cells	hNSCs
**CTG**	0.104	0.0163
**CTF**	>100	>100
95% Cl (from CTG)	0.0931–0.118	0.0148–0.0181
**IC50 (µM) Values of MTX**		
**Assay**	Vero cells	hNSCs
**Immunofluorescence**	0.245	0.334
95% Cl	0.0908–0.463	0.1203–1.187

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
