# Peer review of "Mechanism of Action of Methotrexate Against Zika Virus"

_viruses, 2019, doi:10.3390/v11040338_

Reviewer 1 Report

Acceptable

Reviewer 2 Report

The authors have sucessfully adressed all my concerns and comments. I recommend the study for publication in Viruses. 

Reviewer 3 Report

The revised manuscript is acceptable for pub,ication

Round  1

Reviewer 1 Report

Authors attempt to develop an effective antiviral against ZIKV, the manuscript is well-written, the data are clearly presented and the conclusion is supported by the experiments results. However, given the ZIKV infection causes major problem in the transmitted fetuses, how signficant the MTX is in application? 

Fig. 1A: the authors need to have even exposure of fluorescence, especially in the "nuclei" group (the nuclei of MTX treated group was evidently exposed to fluorescence for a much shorter time.). If DAPI was used, please recover its blue color.

Figure 2 and Table 1: the title of both 3.2 and the legend is "cell cytotoxicity and antiviral efficacy of MTX", I don't see any results to show the "antiviral efficacy of MTX", please fix it.

Figure 3: the authors are encouraged to show an IFA (immunofluorescence assay) as shown in Fig.1A; I think the authors forgot to determine the best anti-ZIKV concentration of the MTX.

Author Response

Authors attempt to develop an effective antiviral against ZIKV, the manuscript is well-written, the data are clearly presented and the conclusion is supported by the experiments results. However, given the ZIKV infection causes major problem in the transmitted fetuses, how signficant the MTX is in application?
We agree with the reviewer that usage of MTX would not be appropriate for treatment of pregnant women with ZIKV infection. However, usage for immunocompromised patients with severe disease who would not be eligible for other treatment options would be one example of a potential application of MTX in a clinical setting. Also, this study acts as a proof-of-principal that the folate pathway can be targeted duringZIKV infection. Other targets within this pathway might also be exploitable for ZIKV drug development, and toxicity could be mitigated through co-administration of metabolites with the drug, such as thymidine with MTX. These points are noted in the text (lines 285-298).

Fig. 1A: the authors need to have even exposure of fluorescence, especially in the "nuclei" group (the nuclei of MTX treated group was evidently exposed to fluorescence for a much shorter time.). If DAPI was used, please recover its blue color.
We acquire all of our images using an ImageXpress Micro automated microscope at the same time from a single plate using the same exposure intensity. Any differences observed in intensity are the result of staining differences between wells; we apply equal intensity changes across all wells for image preparation in accordance with best practices in microscopy. As a result, we cannot adjust the intensity for individuals wells to normalize their appearance. SYTOX green was used to stain the cell nuclei rather than DAPI.
Figure 2 and Table 1: the title of both 3.2 and the legend is "cell cytotoxicity and antiviral efficacy of MTX", I don't see any results to show the "antiviral efficacy of MTX", please fix it.

The authors apologize for this oversight and have included this activity data in Table 1, and have amended the title of Figure 2 to: Determination of cytotoxicity of MTX.
Figure 3: the authors are encouraged to show an IFA (immunofluorescence assay) as shown in Fig.1A; I think the authors forgot to determine the best anti-ZIKV concentration of the MTX.
We have performed this experiment as requested and generated EC50 data for both Vero and hNSCs. These results are presented in Table 1.

Reviewer 2 Report

One major concern is that this manuscript has similar results and experimental design from an earlier dengue virus paper published in JVI.
‘Flaviviruses Are Sensitive to Inhibition of Thymidine Synthesis Pathways
Fischer et al 2013’ referenced as (ref 18) in this manuscript.  The major difference between the earlier paper and this manuscript is the use of Zika virus instead of dengue virus. The results obtained from both studies have significant similarities and have similar conclusions. Therefore it appears that this manuscript is lessened in its impact. However, there have been series of recent publications where drug studies from other pathogens have been tested on Zika virus and found promising. This study therefore deserves consideration. Additionally authors have tested a few more downstream products in the rescue experiments and found to corroborate the role of DHFR as a target for antiviral pathways.
There are a few typos and errors in this manuscript.
Specific comments
Line 38: Culex mosquitoes transmit WNV
45: Rephrase ‘number of hit molecules against ZIKV’
Figure1) It appears that the number of cells are lower in nuclei compared to
Of MTX compared to control and untreated virus infected cells. It may be good to show that the reduction in virus titer is not due to the reduction in number of cells caused by cytotoxicity.
223: ‘was were’
225: Not clear what is defined by ‘Such neurotropism..’
Additionally, it is not clear what the authors are trying explain here: ‘MTX
influences the folate pathway may explain that one of these consequences is a neural tube defect, i.e. microencephaly in infants’. 

Clarify the following statement ‘Although MTX reduced ZIKV titer about ten-fold at 48h PI, MTX could not continuously suppress the ZIKV replication after 48h PI’. 

268-270: How much of the reduction in virus titer is due to cytotoxicity? Mentioned in the following sentence?
‘Accordingly, CC50 measurement with CTG reagent also resulted in high cytotoxicity in the two host cells; however, such toxicity could be misleading because the drug mechanism directly interferes with ATP levels, i.e. the readout measurement of the assay’
Line 280: Add reference as there is a direct mention of ‘As an RNA virus, DENV2 does not directly incorporate deoxythymidine into its genome,” is reported in Fisher et al 2013.
Author Response

Specific comments Line 38: Culex mosquitoes transmit WNV

The authors have corrected the manuscript at this position (line 38).
45: Rephrase ‘number of hit molecules against ZIKV’
This section has been changed to read “several small molecules active against ZIKV” for clarity.
Figure1) It appears that the number of cells are lower in nuclei compared to Of MTX compared to control and untreated virus infected cells. It may be good to show that the reduction in virus titer is not due to the reduction in number of cells caused by cytotoxicity.

In order to clarify this point, the authors used an automated image analysis module to quantify the intensity of virus staining per cell. The results show that the amount of virus per cell is dramatically lower in MTX-treated cells when compared to DMSO controls. We have added this quantification of Figure 1 as data in Table 1.

223: ‘was were’
This has been corrected in the text.
225: Not clear what is defined by ‘Such neurotropism..’ Additionally, it is not clear what the authors are trying explain here: ‘MTX influences the folate pathway may explain that one of these consequences is a neural tube defect, i.e. microencephaly in infants’.
The authors agree that this sentence is not clear and have changed it to the following: It is well known that folic acid supplementation during pregnancy reduces the risk of neural tube defects and other neurodevelopmental complications of pregnancy. It is therefore interesting that inhibition of this pathway prevents ZIKV replication, suggesting that the neurodevelopmental effects of ZIKV and folic acid deficiency may proceed through similar mechanisms.
Clarify the following statement ‘Although MTX reduced ZIKV titer about ten-fold at 48h PI, MTX could not continuously suppress the ZIKV replication after 48h PI’.
This sentence has been changed to: Although MTX was able to reduce ZIKV titers about ten-fold for 48h PI, viral levels began to increase past this time point.
268-270: How much of the reduction in virus titer is due to cytotoxicity? Mentioned in the following sentence? ‘Accordingly, CC50 measurement with CTG reagent also resulted in high cytotoxicity in the two host cells; however, such toxicity could be misleading because the drug mechanism directly interferes with ATP levels, i.e. the readout measurement of the assay’
We address this through quantification of the amount of virus staining per cell in Figure 1 (displayed as EC50 data in Table 1), and have added this sentence at line 277 to highlight this: Furthermore, direct quantification of viral stain per cell revealed a substantial decrease in viral particles per cell for the MTXtreated groups compared to DMSO vehicle controls (Figure 1).
Line 280: Add reference as there is a direct mention of ‘As an RNA virus, DENV2 does not directly incorporate deoxythymidine into its genome,” is reported in Fisher et al 2013.
This has been added to the text.

Reviewer 3 Report

In this study, Beck and colleagues characterize the impact of methotrexate (MTX) treatment on Zika virus (ZIKV) and cell viability/proliferation in cell culture. MTX was identified as a potential anti-ZIKV drug in previous drug repurposing screening studies. Here, they confirm that ZIKV infectious particle production is inhibited by MTX not only in the model Vero cell line but also in human neural stem cells (hNSC), the physiological cell target of ZIKV. The authors further characterize the mode-of-action of MTX by showing that ZIKV replication can be rescued if cells are treated with the downstream products of the DHFR pathway, the target of MTX. All the phenotypes correlate with parallel variations in cell viability when using an ATP-dependent luminescent assay (Cell Titer Glo).

Repurposing FDA-approved drugs such as MTX for potential anti-ZIKV treatments shows great therapeutic potential. Hence, better understanding the mode-of-action of these new antivirals is highly relevant. The experiments demonstrating that MTX effects are specific to the DHFR pathway are convincing (Figs 3 and 4).

However, in light of the results shown in this study, I have concerns regarding the fact that MTX can be actually considered as an antiviral specifically targeting ZIKV. Firstly, to better understand the antiviral mode-of-action of MTX, it is important to identify which step(s) of ZIKV life cycle is (are) impacted by ZIKV. Since adenosine (but not thymidine) treatment can rescue the inhibition of ZIKV infectious titers by MTX, the authors imply in the discussion that viral RNA replication is the target of MTX. Which of viral entry, translation, replication and assembly is inhibited by MTX needs to be firmly addressed.

More importantly, in my opinion, the authors failed to clearly demonstrate that all the ZIKV phenotypes are not simply caused by MTX-induced cytotoxicity or reduced proliferation and hence, indirect. In figure 2B, the authors show that a 48h treatment with 0.5uM MTX reduces the amount of viable cells by more than 50%. In all other phenotypic assays (all other figures), a concentration of 5 uM of MTX is used instead. In these conditions, it is likely that cell proliferation is drastically impacted by such treatment. Considering this, the 1xLog10 decrease in ZIKV production observed upon 5uM MTX treatment may very well be caused by the fact that much less virus-producing cells are viable and not because of a direct antiviral activity. Moreover, if ATP level measurement is not a good readout for cell viability because it is impacted by the DHFR pathway, a cell viability assay more reliable than the CellTiter-Glo (such as LDH or CTF assays or proliferation curves) should be performed instead.  

Additional comments:

- Fig 1: A clear reduction in ZIKV-specific green signal is observed. But this is also the case for the nucleus staining. Can the authors quantify the signal intensity per cell and/or the % of infection? This might give hints whether viral spread or intracellular replication (or both) is impacted by MTX treatment.

-In fig 1D, the authors should include a graph bar for the reference DMSO control.

- Table 1: What is the EC50 of MTX in both Vero cells and hNSCs?

- Fig 3: The authors should include the “leucovirin alone” control condition to ensure that the rescue phenotype observed with the MTX/leucovirin co-treatment is not simply to an unspecific additive effect in case leucovirin alone stimulates replication. The same apply in Fig 4 with the “GAT alone” and “adenosine alone” (without MTX) conditions.

Author Response

Comments and Suggestions for Authors
In this study, Beck and colleagues characterize the impact of methotrexate (MTX) treatment on Zika virus (ZIKV) and cell viability/proliferation in cell culture. MTX was identified as a potential anti-ZIKV drug in previous drug repurposing screening studies. Here, they confirm that ZIKV infectious particle production is inhibited by MTX not only in the model Vero cell line but also in human neural stem cells (hNSC), the physiological cell target of ZIKV. The authors further characterize the mode-of-action of MTX by showing that ZIKV replication can be rescued if cells are treated with the downstream products of the DHFR pathway, the target of MTX. All the phenotypes correlate with parallel variations in cell viability when
using an ATP-dependent luminescent assay (Cell Titer Glo). Repurposing FDA-approved drugs such as MTX for potential anti-ZIKV treatments shows great therapeutic potential. Hence, better understanding the mode-of-action of these new antivirals is highly relevant. The experiments demonstrating that MTX effects are specific to the DHFR pathway are convincing (Figs 3 and 4).
However, in light of the results shown in this study, I have concerns regarding the fact that MTX can be actually considered as an antiviral specifically targeting ZIKV. Firstly, to better understand the antiviral mode-of-action of MTX, it is important to identify which step(s) of ZIKV life cycle is (are) impacted by ZIKV. Since adenosine (but not thymidine) treatment can rescue the inhibition of ZIKV infectious titers by MTX, the authors imply in the discussion that viral RNA replication is the target of MTX. Which of viral entry, translation, replication and assembly is inhibited by MTX needs to be firmly addressed.
We appreciate the reviewer’s comment. It is well known that MTX reduces general nucleotide pools as a mechanism of action. Based on our additional data that show that MTX addition to assay wells with high amounts of cells (30,000) show no effect on cell viability, and that virus titers are suppressed for the first 48 hours PI but begin to recover after this time, we believe this provides evidence to support that MTX inhibits early ZIKV replication at the level of genome synthesis. This notion is supported by complementary studies on related flaviruses (Fischer et al., J. Virol., 2013)

More importantly, in my opinion, the authors failed to clearly demonstrate that all the ZIKV phenotypes are not simply caused by MTX-induced cytotoxicity or reduced proliferation and hence, indirect. In figure 2B, the authors show that a 48h treatment with 0.5uM MTX reduces the amount of viable cells by more than 50%. In all other phenotypic assays (all other figures), a concentration of 5 uM of MTX is used instead. In these conditions, it is likely that cell proliferation is drastically impacted by such treatment. Considering this, the 1xLog10 decrease in ZIKV production observed upon 5uM MTX treatment may very well be caused by the fact that much less virus-producing cells are viable and not because of a direct antiviral activity.
To address this comment, the authors performed the experiment with 5 μM MTX with a higher number of cells seeded (30,000). No observable toxicity from the compound was detected under these conditions. Furthermore, we quantified the average amount of viral antigen per cell in our immunofluorescence in Figure 1 (EC50 data in Table 1), and saw a substantial reduction in virus per cell as the result of MTX treatment as compared to DMSO controls. This suggests that there is a specific antiviral effect that is independent of reduced cellular replication.

We performed a CTF assay as the reviewer requested, and the CC50 of MTX was > 100 μM for the cell lines tested.

Additional comments:
- Fig 1: A clear reduction in ZIKV-specific green signal is observed. But this is also the case for the nucleus staining. Can the authors quantify the signal intensity per cell and/or the % of infection? This might give hints whether viral spread or intracellular replication (or both) is impacted by MTX treatment.

We appreciate the reviewer’s comments and have quantified the average amount of virus antigen present per cell in our immunofluorescence assay using an automated image analysis module to address this point. Shown in Figure 1f, there is a substantial decrease in the amount of viral antigen per cell for MTXtreated sample when compared to DMSO vehicle-treated sample. This would indicate that the reduction in virus production is not only the result of a decrease in the number of cells.

-In fig 1D, the authors should include a graph bar for the reference DMSO control.
This has been added to the figure as requested.
- Table 1: What is the EC50 of MTX in both Vero cells and hNSCs?
These values have been added to Table 1.
- Fig 3: The authors should include the “leucovirin alone” control condition to ensure that the rescue phenotype observed with the MTX/leucovirin co-treatment is not simply to an unspecific additive effect in case leucovirin alone stimulates replication. The same apply in Fig 4 with the “GAT alone” and “adenosine alone” (without MTX) conditions.
These controls have been added to Figure 3 and Figure 4. No effect of these metabolites on cellular replication was observed.